

# 1   Tsunamigenic potential of unstable masses in the Gulf of Pozzuoli,
# 2   Campi Flegrei, Italy

Filippo Zaniboni[1], Luigi Sabino[2], Cesare Angeli[1], Martina Zanetti[1] and Alberto Armigliato[1]
[1] Department of Physics and Astronomy "A. Righi", Alma Mater Studiorum - University of Bologna, Bologna, 40127, Italy
[2] Department of Mathematics and Physics, University of Campania "L. Vanvitelli", Caserta, 81100, Italy
*Correspondence to*: Filippo Zaniboni (filippo.zaniboni@unibo.it)
**Abstract.** Campi Flegrei, one of the most monitored and studied volcanic areas in the world, has recently attracted significant
attention due to the reactivation of its peculiar activity, consisting into small earthquakes, geothermal phenomena and slow
subsidence/rapid uplift cycles, known as bradyseism. While much of the research and of the attention focuses on volcanic
manifestations, the potential hazard posed by gravitational instabilities has received little consideration. The interaction of the
destabilized masses with water can trigger tsunamis, potentially affecting the whole coastline of the Gulf of Pozzuoli, which
lies above the Campi Flegrei caldera. Moving from the limited available geomorphological studies of the area, a set of four
scenarios (three submarine and one subaerial) are here reconstructed. These are simulated through a sequence of numerical
codes, accounting for all the phases of the tsunami process, providing insights into the distribution of tsunami energy and
identifying the most affected coastal stretches. Additionally, the study explores the influence of dispersion effects in the
tsunami propagation and the occurrence of resonance effects in some minor inlets of the Gulf, emphasizing the importance of
accounting for complex and non-linear coastal processes when treating with landslide-generated tsunamis.

## 19   1 Introduction

The recent strengthening of the Campi Flegrei activity has raised many concerns about the potential impacts of volcanic-related
manifestations on local population and infrastructures. The caldera, located in correspondence of an extremely densely
inhabited area in the surroundings of Naples (South Italy), is one of the most active and dangerous in the world. The study of
the hazard correlated to the Campi Flegrei activity, concerning the tephra dispersal in case of eruption, has been already treated
in the scientific literature (Selva et al., 2018). On this basis, a National Civil Protection plan has been realized in case of
emergency (https://mappe.protezionecivile.gov.it/en/risks-maps-and-dashboards/national-planning-phlegraean-fields/). Other
considered potential hazards linked to the eruption of the Campi Flegrei caldera are pyroclastic flows (Neri et al., 2015),
phreatic explosions (Mayer et al., 2016) and mud flows generated by the interaction between the ash ejected from the volcano
and the rain (Isaia et al., 2021). Earthquakes are frequent, though characterized by low magnitude. As an example, according
to the December 2024 INGV - Osservatorio Vesuviano bulletin (https://www.ov.ingv.it/index.php/flegrei-stato-attuale), 273



events have been recorded during the month. Over 93% of these had magnitude lower than 1, four exceeded $M_d = 2.0$ and
the maximum was 3.4.
The interaction of volcanic activity with the sea and the consequent generation of tsunamis received some attention as well:
the work by Paris et al. (2019) tested the effects of submarine volcanic explosions in the Gulf of Pozzuoli by means of
numerical scenarios and a Probabilistic Tsunami Hazard Analysis (PTHA) methodology. Grezio et al. (2020) attempted a
comprehensive approach including earthquakes, submarine landslides and volcanic explosions as potential sources for tsunami
generation affecting the Gulf of Naples, implementing them into a PTHA providing hazard curves for different localities along
the coasts.
In the specific, gravitational collapses are one of the least considered potential sources of natural hazards in the area. Volcanic
activity at Campi Flegrei can induce instability both in the short term, through seismic shaking, and in the long term, due to
slope steepening associated with the caldera uplift. The scenarios considered in Grezio et al. (2020) were purely synthetic,
without evaluation of geological and morphological evidence, of the sliding dynamics and of possible hydrodynamic effects
during the tsunami propagation.
This work aims to bridge to fill these gaps, assessing the tsunamigenic potential of local submarine and subaerial landslides,
evaluating the tsunami distribution patterns and the respective impact on the coasts, at the scale of the Gulf of Pozzuoli basin.
The masses generating the waves are selected based on a worst-case credible approach (see e.g. Zaniboni and Armigliato,
2025), which relies on the present morphology and on the existing knowledge of the mass transport processes in the area. The
tsunami hazard is then assessed through a set of numerical codes, already tested and applied in many other landslide-tsunami
cases. This investigation has been preliminarily performed and reported in Sabino (2024): here it is extended including the role
of dispersion in the tsunami propagation, since it can have significant influence in the impact of the waves on the coasts.
**1.1 Campi Flegrei volcano**
Campi Flegrei is a complex volcanic system with a history spanning at least the last 78,000 years (Perrotta et al., 2006),
characterized by intense unrest episodes involving ground deformation and seismicity (de Natale et al., 2006), together with
explosive eruptions and variable vent locations (Bevilacqua et al., 2015). Seismic studies have revealed a large magmatic sill
at depth, potentially connected to the surface through deep fractures (Zollo et al., 2008). The heart of Campi Flegrei is the vast
caldera, an almost circular structure with a diameter of about 10 km (marked by the red dashed line in Figure 1) involving the
western districts of Naples in its subaerial expression, with a total potentially affected population of more than 360 thousand,
living in the cities of Pozzuoli, Bagnoli and Bacoli. The submarine part of the caldera is covered by the Gulf of Pozzuoli, a
small, shallow water sub-basin of the wider Gulf of Naples. The landscape of this area has been shaped by several volcanic
events: the caldera was formed by a catastrophic volcanic explosion that occurred approximately 39,000 years ago. This event,
an eruption of 100-200 km³ of rock called "Ignimbrite Campana" (Rosi et al., 1983; Perrotta et al. 2006), shaped the region's
topography, creating a unique landscape characterized by hills, fumaroles, and hot springs. At the centre of the caldera rises
the Solfatara crater, a focal point for the volcanic and geothermal activity in the area. Historical reports from Roman times





reveal a general trend of slow subsidence (rate of about 1-2 cm/yr), alternated with occasional episodes of faster uplift (Di Vito
et al., 2016; De Vivo et al., 2020), a peculiar behaviour that took the name of "bradyseism". Soil subsidence can lead to coastal
flooding, as testified by the submerged park of Baia, where Roman houses and constructions are still visible now at 6-8 m
depth. The last significant eruption of Campi Flegrei caldera occurred in 1538, with 0.03 km³ of erupted products that gave
rise to the hill of Monte Nuovo in one single night (De Vivo et al., 2001). After that, the floor of the caldera underwent slow
and regular subsidence. In more recent times, some episodes of uplift interrupted this phase: 74 cm in 1950-52, 159 cm in
1970-72 and 178 cm in 1982-84 (Del Gaudio et al., 2010), resulting in a maximum rise of about 3.5 meters of the ground in
the city of Pozzuoli, with shallow micro-seismicity recorded in response to fluid movement episodes.

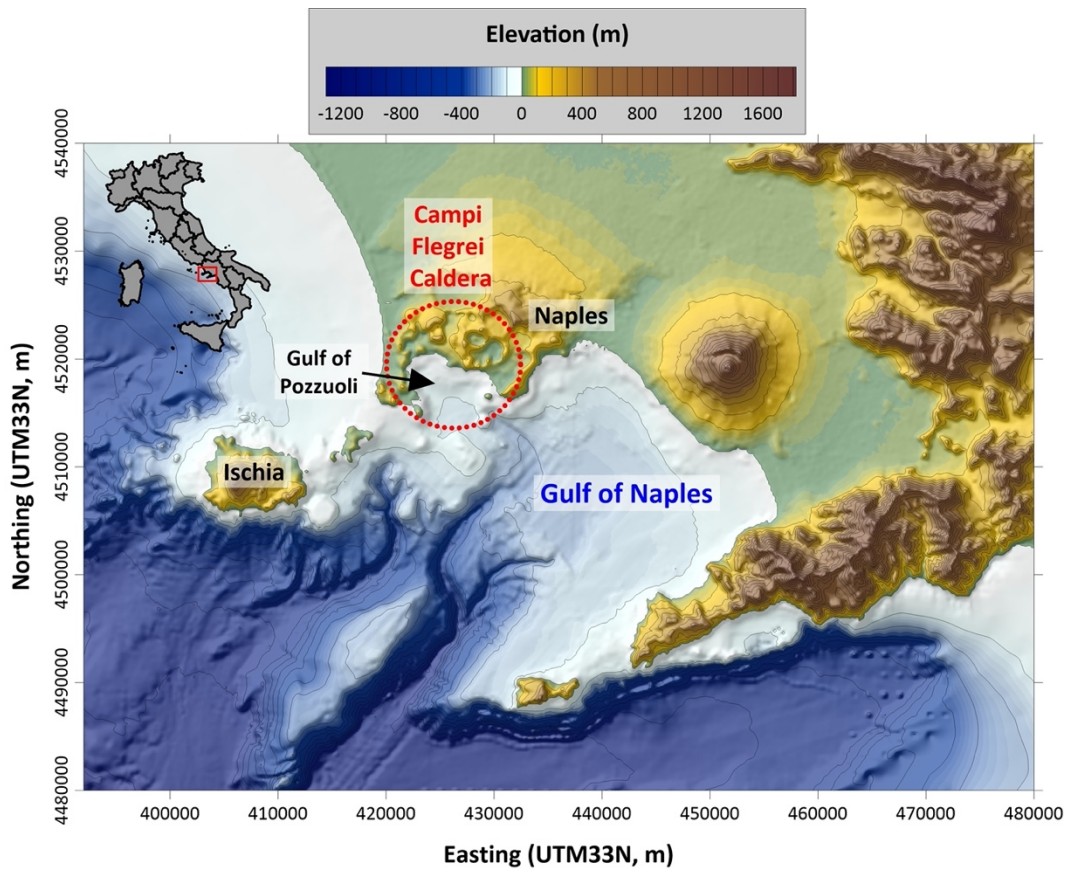



**Figure 1. Morphological map of the Gulf of Naples (South Italy). The red-dashed circle delimits approximately the Campi Flegrei**
**caldera, involving a subaerial part (west of Naples) and a submarine portion (Gulf of Pozzuoli).**
Starting from the second half of 2023, the bradyseism has resumed, with seismic activity intensifying, although most of the
events were characterized by low magnitudes (about 90% of the events had magnitudes less than 1.0, see




https://www.ov.ingv.it/index.php/monitoraggio-e-infrastrutture/bollettini-tutti/bollett-mensili-cf), with maximum depths of
around 4 km, predominantly concentrated within the first 2 km (Danesi et al., 2024). The sequence culminated with the seismic
events on September 27th, 2023 ($M_d = 4.2$) and October 2nd, 2023 ($M_d = 4.0$), located respectively in the area between Bagnoli
and Pozzuoli and in the Pisciarelli-Solfatara area.

## 2 Data and methods

The investigation on the tsunami hazard in the gulf of Pozzuoli associated to the Campi Flegrei activity, triggering potential
instabilities interacting with water, is here performed through numerical methods, which in turn implement approaches that
are described in this section. An initial necessary premise concerns the importance of considering the dispersive effects in
landslide-tsunami simulations, a task that is usually underestimated but whose effects could be relevant in the analysis of the
tsunami impact on the coast.

### 2.1 Landslide-tsunamis and dispersive effects

Tsunamis are oscillations induced by a perturbation of the equilibrium state involving the whole water body, which can be
produced by sudden changes in the sea bottom (earthquakes, submarine landslides, underwater volcanic explosions) or upon
the sea surface (subaerial landslides, atmospheric perturbations, cosmogenic tsunamis).
In general, in hydrodynamics, each wave is subject to the dispersion relation, linking the phase velocity $c$ to the wave number
$k$ (or to its inverse, the wavelength $\lambda$): the smaller is the second (i.e., the bigger is the wavelength), the higher is the first. In
short, longer oscillations are faster than shorter ones (for a general overview about this topic see for example Saito, 2019).
A tsunami can be considered mostly as a "packet" of waves, meaning that it is not a monochromatic wave, but it results from
the superposition of different components, each characterized by a specific wavelength, then with its own velocity. When $\lambda$ is
much bigger than the water depth $h$ of the involved basin, the hydrodynamic equations can be significantly simplified by
means of the shallow-water (SW) approach, in which tsunamis are treated as "long-waves". This is generally valid for waves
generated by earthquakes, since the source has dimensions that is usually much bigger than the typical depth of the sea. When
the tsunami trigger is provided by other phenomena, such as landslides, SW is known to neglect important hydrodynamic
effects, such as dispersion, that can affect significantly the wave propagation and impact on the coast. It is generally agreed
that SW is considered proper when the ratio $\lambda/h$ is bigger than 20, while for lower values (i.e. for "shorter" waves) a more
sophisticated and higher-order version of the hydrodynamic equations should be considered, as for example the Boussinesq
approximation (hereafter accounted for as non-hydrostatic approach, NH).
A quantification of the dispersive effects on the tsunami propagation was attempted in Glimsdal et al. (2013). A rearrangement
of the considerations found in that work leads to the estimation of the ratio between distance and initial signal wavelength for
which the dispersion becomes relevant and cannot be neglected anymore. Calling this (non-dimensional) dispersion distance
as $D$, the expression is:





$$D = 0.025 \cdot \left(\frac{\lambda}{h}\right)^2 \tag{1}$$

For "long waves", then, $\lambda/h \approx 20$ and $D \approx 10$: dispersive effects manifest at least at a distance ten times the initial
wavelength. Table 1 reports some typical values for wavelengths and water basin depths found respectively for: earthquake-
tsunamis (first line), for which generally SW approach is applicable; submarine landslide tsunamis (second line) for which SW
is valid only on limited domains; subaerial landslide tsunamis (third line), which can be considered in general as deep-water
(DW) waves, for which the NH approach is more suitable.
In general, we can see that dispersion affects every tsunami, also the "longer" ones, but it manifests only on very long distances,
where probably the perturbation itself has already been damped by other propagation effects. Neglecting dispersion means that
the different components of the tsunami travel all together and impact the coast at the same time; conversely, when considering
it, the longer components travel much faster than the shorter ones. This provokes an "unpacking" of the tsunami, that is much
more evident and effective with increasing distance from the source: in general, then, neglecting dispersion causes an
overestimation of the tsunami effects.
**Table 1. Examples of waves and of computing of respective dispersion distance. $\lambda$: initial wavelength; $h$: water depth; $D$: dispersion**
**distance, computed by Eq. (1); $d$: distance for which dispersion effects become predominant ($d = D \cdot \lambda$).**

| $\lambda$ (km) | $h$ (m) | $\lambda/h$ | Wave type | $D$ | $d$ (km) |
|---|---|---|---|---|---|
| 100 | 4000 | 25 | SW | 15.6 | 1562.5 |
| 5 | 500 | 10 | Weakly SW | 2.5 | 12.5 |
| 1 | 200 | 5 | DW | 0.625 | 0.625 |

**2.2 Overview on numerical techniques**
Landslide-generated tsunamis are highly complex processes. Assessing their impact on the coast through numerical
simulations requires numerous approximations and assumptions. The approach adopted here divides the entire process into
distinct phases: i) the landslide motion, ii) the transfer of energy from the mass to the water, iii) tsunami propagation, and iv)
the wave's impact on the coast. Each phase is treated independently, with no back-interactions considered (for example, the
mass-water interaction, where the wave's propagation can alter the water column and influence the dynamics of the landslide
on the seafloor, is neglected in this study). The phases are addressed sequentially, with the output of one stage serving as the
input for the next.
*Landslide motion*. The movement of the mass along the seafloor triggers the tsunami. Unlike earthquake-generated tsunamis,
where the source process can be considered instantaneous relative to the wave's propagation (at least as a first approximation),
landslide-tsunamis involve a finite time for the generation phase. Thus, it is essential to accurately describe the mass dynamics
and the evolution of shape changes. This is achieved using the numerical code UBO-BLOCK (see detailed description in Tinti



et al., 1997), which divides the sliding mass into interacting portions. The equation governing the centre of mass (CoM) motion
is determined by the forces acting on the mass: gravity, buoyancy, basal friction, surface drag, and an internal interaction force
accounting for energy dissipation due to deformation. The resulting time-series of geometric and dynamic quantities describing
the landslide motion is then generated. Applications of UBO-BLOCK to landslide-tsunami events can be retrieved in
Tryantafillou et al. (2020), Zaniboni et al. (2021), Gallotti et al. (2021), Gasperini et al. (2022), Gallotti et al. (2023), Zaniboni
et al. (2024) and in the references therein.
*Tsunamigenic impulse*. The next step involves assessing the perturbation to the water column caused by the sliding motion
along the seafloor. This disturbance provides the dynamic forcing term for the wave propagation equations, which are not
instantaneous but evolve over time. The change in the seafloor due to the landslide is interpolated onto the tsunami grid nodes,
with a filter applied to local sea depths that suppresses high frequencies. These tasks are handled by the intermediate code
UBO-TSUIMP (details in Tinti et al., 2006).
*Tsunami propagation and coastal impact*. The final two phases are simulated using the JAGURS code, which solves the fluid
dynamics equations using finite difference methods. Tsunami propagation can be modelled using either the shallow-water
(SW) equations or a more sophisticated approach that accounts for vertical variations in hydrodynamic quantities,
implementing for example the Boussinesq model (NH). The non-hydrostatic method allows the code to capture dispersion
effects, which are crucial for landslide-generated tsunamis, as described in Section 1.1. JAGURS supports simulations over
computational grids with varying resolutions (using a nested grid approach) and can model coastal inundation. It is also
optimized for parallel computing with MPI and OpenMPI coding, enabling the simulation of dispersive effects over large
computational domains (see Baba et al., 2015, for further description). While JAGURS is widely used for earthquake-generated
tsunamis (Ren et al., 2021; Ehara et al., 2023), for landslide-generated events it requires some adaptations, due to the nature
of the phenomenon itself. As previously mentioned, the tsunamigenic impulse for landslide-tsunamis is not instantaneous:
then, it must be provided to the code as a sequence of single impulses, one for each time step describing the landslide motion
along the sea bottom and producing the perturbation.

**2.3 Computational grid**

The numerical codes applied here require as input a regularly spaced computational grid, whose definition and assembling
needs a compromise among different aspects: the detailed description of the morphology, mainly for the coastal areas, requires
a huge number of nodes, that on the contrary needs to be limited basing on the computational resources available. Additionally,
landslide-tsunamis are known to affect limited domains, due to the dispersive effects characterizing their oscillations that
produce a rapid damping of their amplitude. In the specific case studied here, the morphology of the seabed suggests that the
mass instabilities induced by the Campi Flegrei activity have typically small volumes, producing waves that will presumably
travel modest distances, with reduced consequences on the coasts. Under all these considerations, the selected tsunami
computational domain covers the Pozzuoli Gulf for an area of 12 km x 11 km approximately, with a spatial step of 20 m and



a total number of nodes of about 340 thousand (see Table 2). Raw data have been retrieved from the database MaGIC for the
coastline and the bathymetry (DPC), and from the database Tinitaly (Tarquini et al., 2023) as concerns the topography.
The morphology represented in Figure 2 shows some remarkable features as concerns landslide-tsunami triggering and
propagation: i) the underwater slopes are in general quite gentle, inhibiting high initial acceleration of the sliding masses, one
of the most important factors in tsunami genesis (e.g. Lovholt et al., 2015). Only the areas close to the Gulf's mouth opposite
sides, i.e. Capo Miseno on the west and Nisida Bank on the east (see Figure 2 for toponyms), show steep submarine gradients.
ii) the coastal slope is generally flat, favoring water ingression and tsunami penetration; only the two abovementioned areas
(Capo Miseno and Nisida) are characterized by steep coastal profiles, that can also produce rapidly moving collapses into the
water. iii) the sea is rather shallow within the Gulf, with maximum depth reaching about 100 m, while it deepens eastward and
southward. This will have consequences on the wave propagation in the basin, since, as we have seen, the sea depth deeply
influences the tsunami behaviour during the propagation. iv) finally, but not less importantly, at this level of detail it is possible
to represent the main piers and harbour structures (well visible at Pozzuoli and Bagnoli, Figure 2), which influence the wave
propagation with local effects, such as for example multiple reflections and resonance.

**Table 2. Parameters of the tsunami computational domain covering the Gulf of Pozzuoli:**
$N_x$, **number of nodes in the west-east direction;** $N_y$, **number of nodes in the north-south direction;** $\Delta x, \Delta y$ **spatial grid step along the**
$x$ **and** $y$ **directions respectively;** $N_{TOT}$, **total number of nodes;** $h_{max}$ **maximum water depth in the computational grid.**

| $N_x$ | $N_y$ | $\Delta x, \Delta y$ (m) | $N_{TOT}$ | $h_{max}$ (m) |
|---|---|---|---|---|
| 601 | 576 | 20 | 346 176 | 169 |







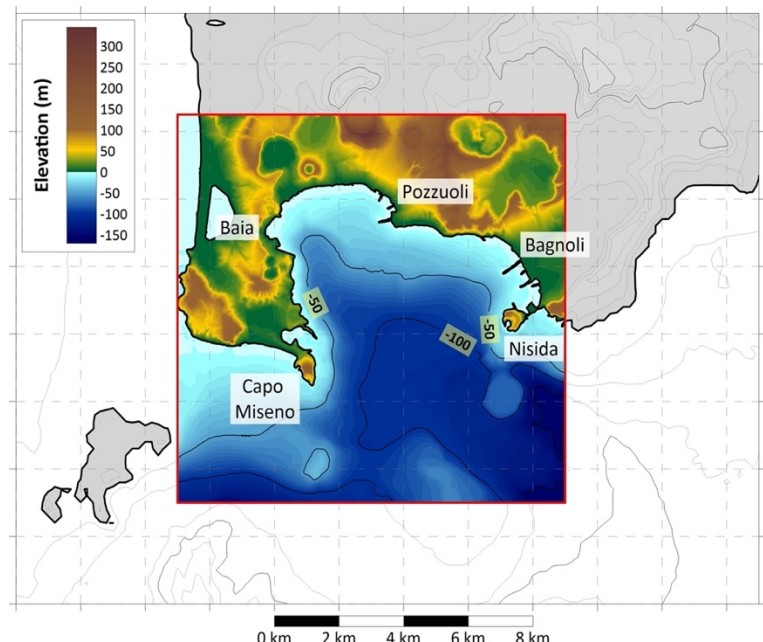


**Figure 2. Computational grid for the tsunami simulation, covering the Gulf of Pozzuoli with a spatial step of 20 m.**
**3. Results**
The evaluation of the landslide-tsunami hazard in the Gulf of Pozzuoli requires, as a first step, to identify the sources, i.e.
masses that have the potential to generate waves, both in the submarine and in the subaerial environments. This investigation
is here performed through a scenario approach, meaning that the sources are hypothesized starting from geological and
morphological evidence, studying their signatures (scars, slopes, deposits) and the ensuing tsunamis evaluated through the
numerical routine previously illustrated. The assumption is that such scenarios represent the range of possible and credible
worst events occurring in the area, following the approach called Worst-case Credible Tsunami Hazard Assessment (WCTHA,
Zaniboni and Armigliato, 2025), which is suitable to phenomena like landslide-tsunamis, where neither recurrence time are
defined nor extensive catalogues of events are available, contrary to the most adopted approach in tsunami science (PTHA,
see e.g. Behrens et al., 2021).
In the following, as a first step, the search for mass collapse traces in the Gulf of Pozzuoli within the scientific literature is
reported; from this, a set of numerical scenarios is defined and the numerical routine previously described applied, providing
a quantification of the hazard associated to the generated tsunamis.
**3.1 Mass instabilities in the Gulf of Pozzuoli and landslide scenarios**
Recent magnetic surveys in the Gulf of Naples evidenced the presence of consistent deposits on the seabed, testifying an
intense mass transport activity in the whole area (de Ritis et al., 2024), that could have probably generated tsunamis affecting





also the Gulf of Pozzuoli. The focus, though, is here centred on local sources, i.e. potential landslides triggered by the seismic
shaking related to the Campi Flegrei caldera, then along the slopes (submarine and subaerial) in the Gulf of Pozzuoli area.
As already mentioned, this is a small, shallow water basin, whose detailed stratigraphic and morphological features have been
revealed by high-resolution seismic and bathymetric surveys (Aiello et al., 2012; Aiello et al., 2016; Somma et al., 2016). The
area is characterized by numerous seismic units, both volcanic and sedimentary, and by a complex tectonic setting related to
the intense Campi Flegrei volcanic activity. The northern part of the basin shows an inner shelf with maximum depth of 50 m,
deepening to 100 m with gentle slopes and delimited by a belt of submarine volcanic edifices: starting from East, they are
called Miseno Bank, Pentapalummo Bank and Nisida Bank (see Figure 3A). The morphological investigations have revealed
the presence of deposits in the deeper and flat part of the Gulf, especially around Miseno Bank and in the central part of the
basin (areas in light brown, Figure 3A; Aiello et al., 2012). The seismic sections reported in the same publication evidence the
presence of buried deposits with peculiar characteristics, which have been characterized and denoted as paleo-landslides.
Figure 3B reports the profile along which they have been found, evidenced in red: one lies along the transect L69_07, extending
in the E-W direction and showing a potential deposit close to Nisida; the second refers to the line L74_07, running in the N-S
direction from Pozzuoli. The morphological evidence close to Miseno Bank and the two buried deposits found in the seismic
profiles will be taken as the basis for the submarine landslide scenarios adopted for this investigation.

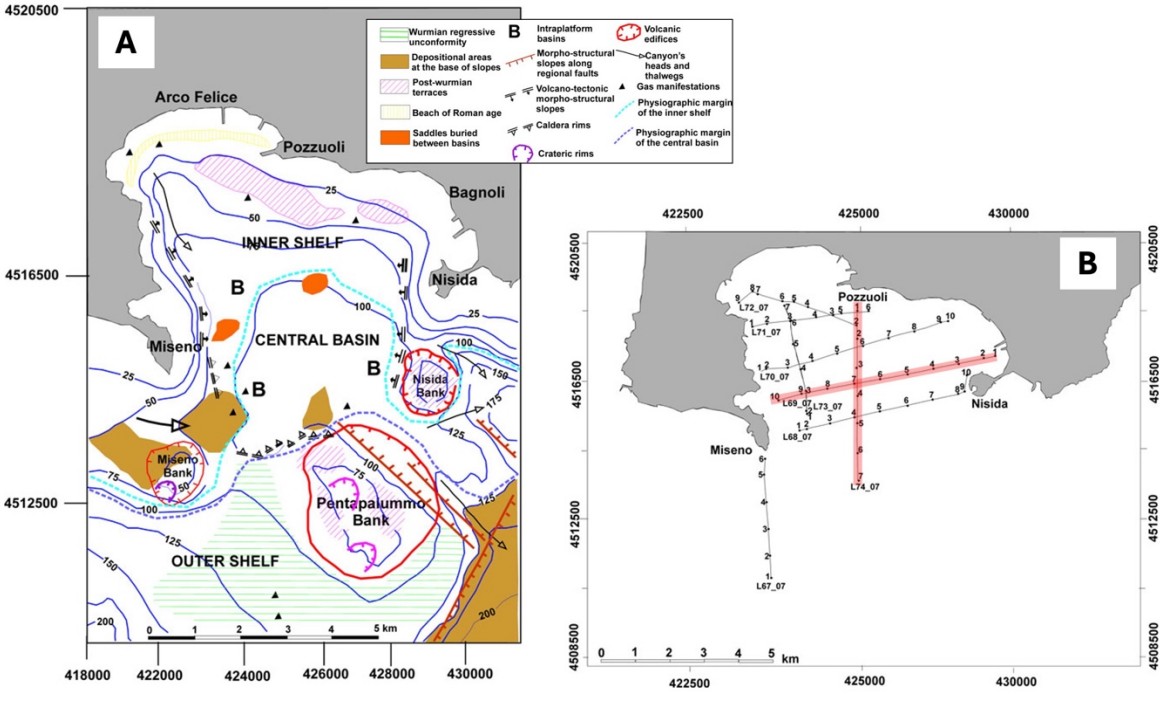




**Figure 3. Panel A) Sketch of the main morphological submarine features of the Gulf of Pozzuoli. Panel B) seismic lines acquired in**
**the seismic survey illustrated in Aiello et al. (2012); the profiles with traces of paleo-landslides are evidenced in red. Figures modified**
**from Aiello et al. (2012).**
As concerns subaerial sliding, a comprehensive geodatabase (CAFLAG) documents 2302 landslide events from 1828 to 2017,
most of which consists of rock falls affecting volcanic slopes and rainfall-induced slides in pyroclastic deposits (Esposito and
Matano, 2023). Landslide hazards affect over 15% of the subaerial Campi Flegrei area, with varying risk levels among towns
(Calcaterra and di Martire, 2022). The events of interest here are the ones potentially interacting with water: the candidates are
then restricted to collapses along the coastline that are not simple rockfalls (surely generating high waves but only locally,
dissipating quickly), but more complex landslides involving a coherent mass impacting the water and moving along the
seafloor.
Based on all these considerations, a set of four landslide scenarios has been arranged for the numerical simulations.
Table 3 summarizes their geomorphological characteristics (volume, area, maximum thickness and initial elevation),
while Figure 4 reports the position of the initial masses for each case. Three of them are submarine and cover different
positions over the Gulf of Pozzuoli, while the fourth is subaerial.
Scenario 1 (blue contour in Figure 4) is located just south of Capo Miseno, at the western end of the Gulf, and has been
reconstructed starting from the deposit shown in brown in Figure 3. The hypothesis is that the mass detached from the
seafloor depression between Capo Miseno and the Miseno Bank, with the constraint to obtain a volume comparable to
the observed deposit at the toe of the slope: since an accurate estimate of this does not exist, a conservative approach
has been adopted, assuming a volume of almost 3 million m³ for this scenario.
Scenario 2 (in magenta, Figure 4) lies in the central part of the Gulf and has been built based on the buried paleo-
landslide recognized in the seismic profile just south of Pozzuoli (Figure 3). Its morphology is quite similar to Scenario
1, with volume and initial area slightly smaller, though its shape is more elongated in the sliding direction.
Scenario 3 (green boundary, Figure 4) is placed in the eastern part of the basin, and recalls a buried paleo-landslide as
well, relative to another seismic transect (E-W from Nisida, see Figure 3). This scenario shows a slightly larger volume
(around 4 million m³), a slightly larger initial thickness and moves from shallower water: all these elements suggest a
higher capability of triggering relevant waves.
Scenario 4 (in black, Figure 4) is the only subaerial scenario, and is located along the coastal cliffs of Capo Miseno, at
the western end of the Gulf of Pozzuoli. It has been chosen by morphological considerations, assuming the large coastal
subaerial scar of the eastern flank of Capo Miseno - still visible now - originated from a single, sudden collapse. The
resulting scenario is quite different from the other cases: volume much smaller (around half million m³) and much
larger initial thickness (maximum of more than 50 m).
**Table 3. Morphological characteristics of the Gulf of Pozzuoli landslide scenarios.**





| Scenario | Environment | Volume ($10^6$ m$^3$) | Area (km$^2$) | Maximum Thickness (m) | Initial maximum elevation (m) |
|---|---|---|---|---|---|
| 1 | Submarine | 2.98 | 0.77 | 9.47 | -49 |
| 2 | Submarine | 2.30 | 0.56 | 10.11 | -37 |
| 3 | Submarine | 4.12 | 0.63 | 12.73 | -27 |
| 4 | Subaerial | 0.58 | 0.03 | 56.77 | 126 |

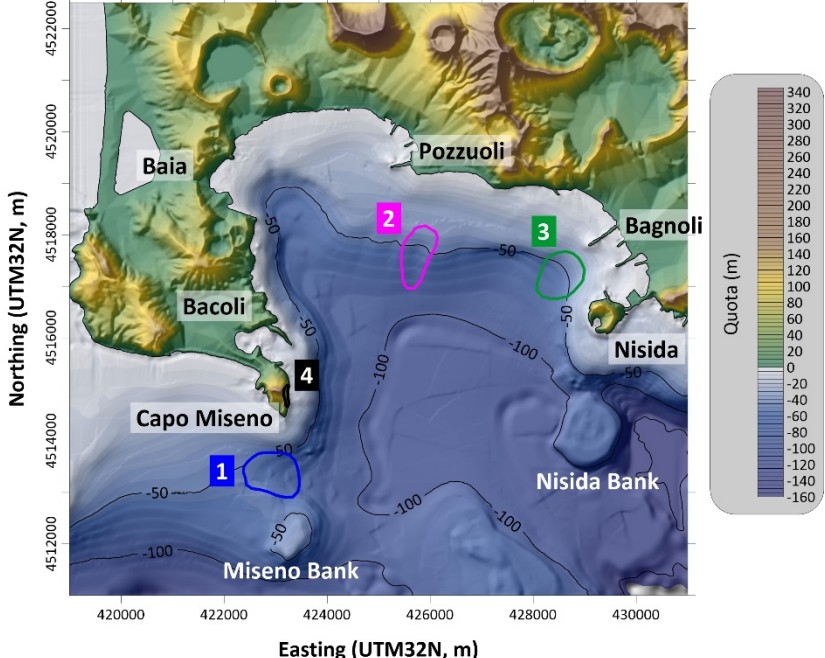

**Figure 4. Location of the initial bodies for the four landslide scenarios hypothesized: in blue, Scenario 1, between Capo Miseno and Miseno Bank; in magenta, Scenario 2, south of Pozzuoli; in green, Scenario 3, offshore Bagnoli; in black, Scenario 4, along the cliffs of Capo Miseno.**

## 3.2 Landslide simulations

The code UBO-BLOCK has been applied to each of the scenarios previously described. The software requires as input the initial landslide configuration (the undisturbed sliding surface and the upper surface of the mass), the predefined trajectory of the CoM and the lateral boundaries of the surface swept by the sliding motion. In this way it is possible to obtain the time history of the landslide shape changes and of its dynamics, representing the input for the computation of the tsunamigenic impulse.





### 3.2.1 Scenario 1

The first scenario is submarine, with a volume around 3 million m³, and is placed south of Capo Miseno. Figure 5A shows the initial thickness distribution (yellow-red scale), and the predefined trajectory for the CoM, that has been determined based on the position of the final deposit found from the geomorphological survey (dashed-blue boundary). Figure 5B reports a section of the landslide, taken along the CoM trajectory: as previously noticed, the slopes are quite gentle, with an average value of 1.5° for this case. In the simulation, the sliding mass settles between 80 and 100 m sea depth (red line, Figure 5B), reaching a maximum velocity of around 5 m/s after 200 seconds (Figure 5C), and decelerating quickly due to low seafloor gradient.

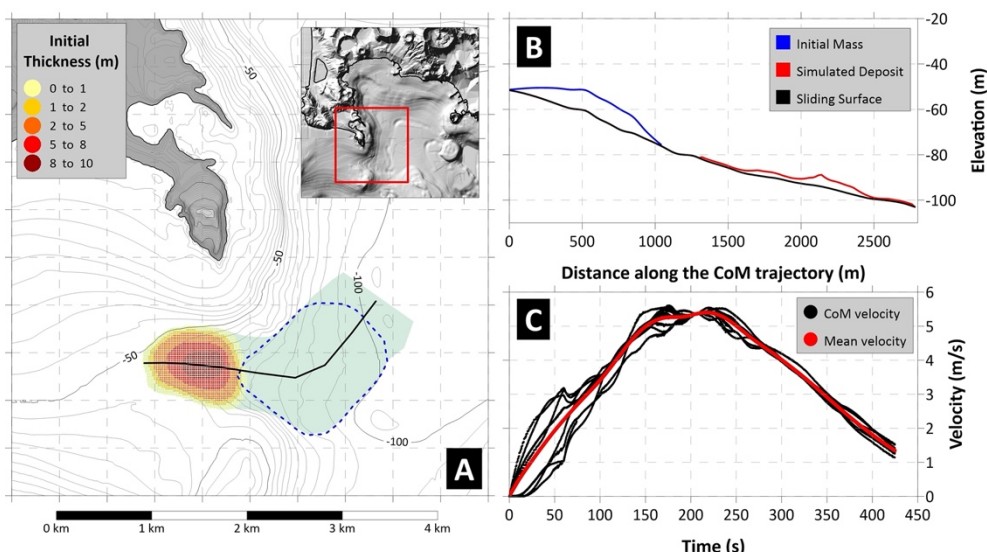

**Figure 5. Panel A) Map of the initial sliding mass for Scenario 1, with initial thickness shown by the yellow-red scale. The black line marks the CoM predefined trajectory, defined to fit the observed deposit (dashed-blue boundary). The area swept by the sliding motion is highlighted in green. Panel B) Landslide profile along the CoM trajectory: in black the undisturbed sliding surface; in blue the initial mass; in red the simulated deposit. Panel C) Sliding velocity time history: in red the average, in black the values for each CoM.**

### 3.2.2 Scenario 2

The second scenario is still submarine and is placed in the central part of the Gulf of Pozzuoli, about 1 km offshore from the piers of the homonymous city (Figure 6A). The hypothesized initial mass has been placed along the steeper slope connecting the shallow-water shelf to the deeper sea. The volume is similar to Scenario 1 (see Table 2), and the predefined sliding direction (black line in Figure 6A) mainly extends in the north-south direction. The simulation shows the deposit reaching the sub-horizontal seafloor at about 100 m depth (Figure 6B), with acceleration and deceleration phases almost symmetric, around the peak velocity value of 9 m/s (Figure 6C).





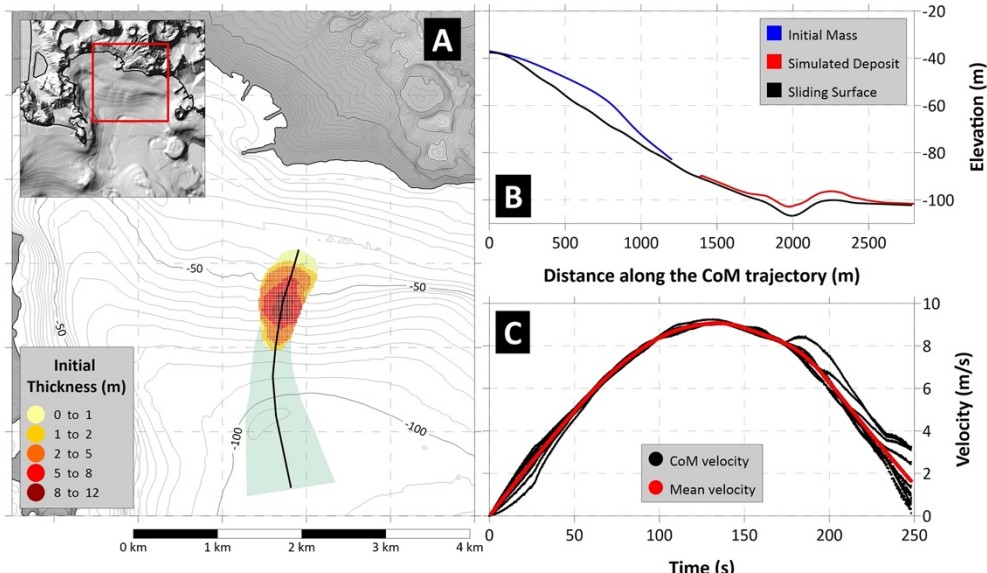

**Figure 6. Panel A) Map of the initial sliding mass for Scenario 2, with initial thickness shown by the yellow-red scale. The black line marks the CoM predefined trajectory. The area swept by the sliding motion is highlighted in green. Panel B) Landslide profile along the CoM trajectory: in black the undisturbed sliding surface; in blue the initial mass; in red the simulated deposit. Panel C) Sliding velocity time history: in red the average, in black the values for each CoM.**

### 3.2.3 Scenario 3

The last submarine scenario is located on the eastern side of the basin, just north of the small peninsula of Nisida and few hundreds of meters offshore Bagnoli (Figure 7A). As for the previous cases, the initial thickness is shown by the area in the yellow-red scale, showing that the mass is not distributed uniformly, but with the thicker part placed in deeper water. The sliding motion follows the main bathymetric gradient, south-westward (black line, Figure 7A), stopping at about 100 m depth, where the slope is quite horizontal (Figure 7B). The dynamics is characterized by a shorter acceleration phase, reaching the maximum velocity of almost 9 m/s after around 2 minutes (Figure 7C), followed by a longer deceleration taking 4 minutes before stopping.





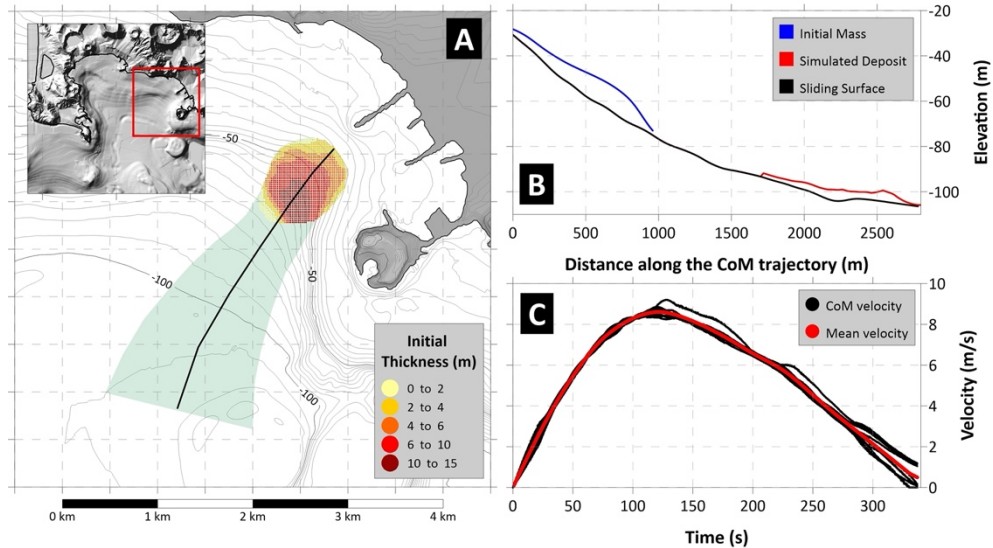

301

**Figure 7. Panel A) Map of the initial sliding mass for Scenario 3, with initial thickness shown by the yellow-red scale. The black line marks the CoM predefined trajectory. The area swept by the sliding motion is highlighted in green. Panel B) Landslide profile along the CoM trajectory: in black the undisturbed sliding surface; in blue the initial mass; in red the simulated deposit. Panel C) Sliding velocity time history: in red the average, in black the values for each CoM.**

### 3.2.4 Scenario 4

This scenario is the only subaerial one and presents very different morphological features compared to the previously illustrated cases: much smaller volume and initial area (see Table 2), much larger initial thickness, and an aspect ratio (length/width) very different from the submarine cases, as visible from Figure 8A. The initial mass has been hypothesized "filling" the subaerial, coastal scar still visible now along the eastern flank of Capo Miseno. Though purely theoretical, this type of collapse is not unusual in the whole area of the Gulf of Naples; this scenario can be considered an endmember for this category of landslides. The simulation shows that the deposit reaches a sub-horizontal area at 70-80 m depth (Figure 8B), with dynamics again different from the previous cases: a sudden acceleration brings the sliding mass at about 23 m/s within 10 seconds (Figure 8C), due to the very steep slope characterizing the first part of the trajectory (black line in Figure 8B); then the submarine shelf in very shallow water provokes an abrupt deceleration, down to 5 m/s, before a second acceleration due to the increasing slope between 20 and 60 m b.s.l., up to 10 m/s. Finally, the slide stops about 2 minutes after the onset.





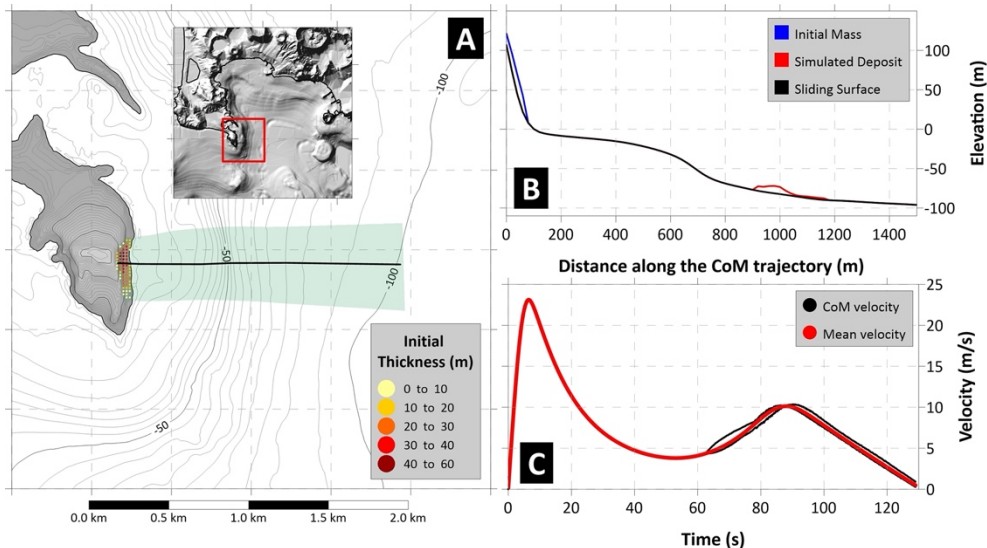

**Figure 8. Panel A) Map of the initial sliding mass for Scenario 4, with initial thickness shown by the yellow-red scale. The black line marks the CoM predefined trajectory. The area swept by the sliding motion is highlighted in green. Panel B) Landslide profile along the CoM trajectory: in black the undisturbed sliding surface; in blue the initial mass; in red the simulated deposit. Panel C) Sliding velocity time history: in red the average, in black the values for each CoM.**

## 3.3 Tsunami simulations

As illustrated previously, tsunamis can be significantly affected, during the propagation, by the hydrodynamic effect of dispersion, due to the different phase velocity characterizing its components. This phenomenon is particularly evident for short oscillations, which are more often induced by landslides. It is possible to estimate the distance at which such effects become predominant through Eq. (1), applying it to each of the studied scenarios. The dispersion can be quantified through the ratio $\lambda/h$, with $\lambda$ tsunami wavelength and $h$ sea depth that can be determined as follows:

- $\lambda$, the initial wavelength of the tsunami, can be assessed in a first approximation as twice the longitudinal length of the slide, called $b$. This assumption, adopted in Heidarzadeh et al. (2023), though quite simplistic and rough, provides a first reasonable indication for this quantity.
- Since the focus here goes to the minimum value for the ratio $\lambda/h$, delimiting the validity of SW approach, a value representing the maximum depth $h$ for the tsunami propagation inside the Gulf of Pozzuoli is assumed: 105 m. The waves that are in the SW regime for this value, will satisfy this requirement also for shallower water.

Table 4 reports the estimations obtained for the landslide scenarios here adopted: the three submarine cases are in the SW regime, with the dispersion manifesting only for distances larger than the Gulf of Pozzuoli dimension. The subaerial case (Scenario 4) generates waves that are suddenly dominated by the dispersive effects: for this case the use of SW for the propagation on the whole Gulf of Pozzuoli seems not suitable.





**Table 4. Computation of the dispersion distance for the four scenarios here studied ($b$ landslide length; $\lambda$: initial wavelength, assumed to coincide with $b$; $h$ water depth (fixed at 105 m); $D$ dispersion distance, computed by Eq. (1); $d$: distance for which dispersion effects become predominant).**

| Scenario | Environment | $b$ (m) | $\lambda/h$ | $D$ | $d$ (km) |
|:---:|:---:|:---:|:---:|:---:|:---:|
| 1 | Submarine | 1090 | 20.7 | 10.7 | 11.7 |
| 2 | Submarine | 1260 | 24.0 | 14.4 | 18.1 |
| 3 | Submarine | 1010 | 19.2 | 9.2 | 9.3 |
| 4 | Subaerial | 170 | 3.24 | 0.26 | 0.05 |

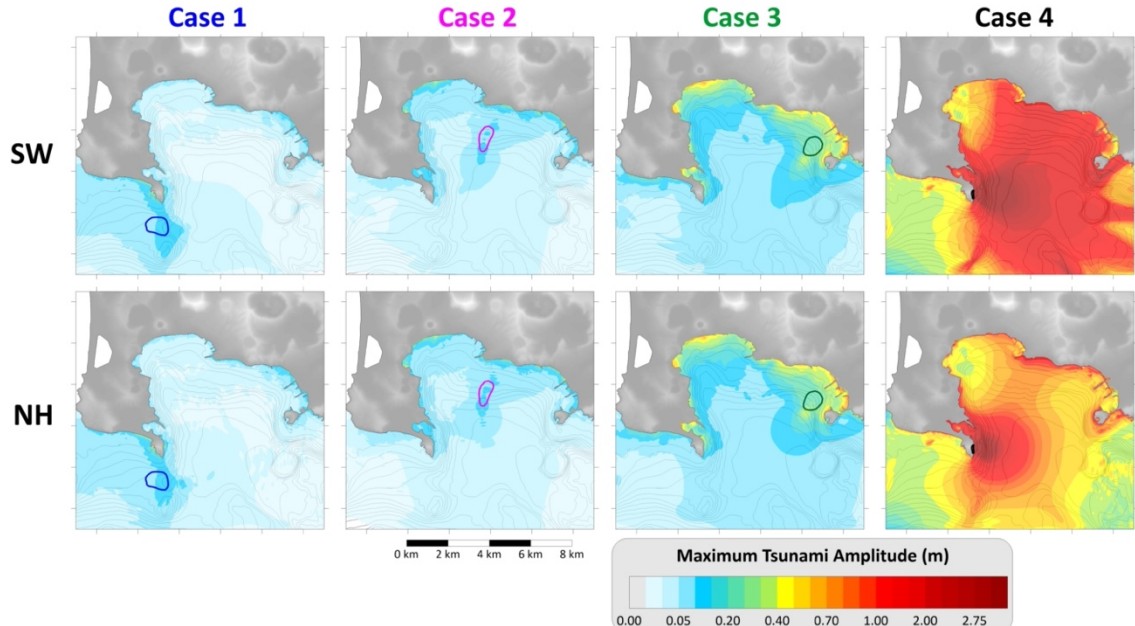

**Figure 9. Maximum water amplitude on each node of the computational grid for the four scenarios considered: each of them are simulated through the shallow-water (SW) and the Boussinesq (NH) approach. The coloured boundaries report the initial position of the respective landslide source.**

The simulations of the tsunamis generated by the sliding scenarios were performed using the JAGURS software, employing both the SW and the non-hydrostatic (NH) approaches, with the non-linear version of the equations in both cases. This strategy allows to investigate the suitability of the considerations previously made on the effects of dispersion on the tsunami propagation. Figure 9 illustrates the maximum water amplitude on each point of the computational domain for each scenario, comparing the two approaches: SW (upper row of plots) and NH (lower row). For the submarine scenarios (1, 2 and 3) the differences are negligible; for Case 4 (last column) significant differences are evident, with the NH approach showing much



more localized effects compared to SW. This confirms the earlier hypothesis that dispersion effects cannot be neglected for
sources of this type.
From a hazard assessment perspective, submarine landslides generate relatively small tsunamis in the Gulf of Pozzuoli. In
Cases 1 and 2, the maximum wave amplitudes are on the order of some tens of centimetres. In Case 3, the maximum amplitude
exceeds 50 cm along the coastal stretch between Bagnoli and Nisida, which, while not catastrophic, could still cause damages
to small boats and generate currents in smaller sub-basins. In contrast, Case 4 produces more significant waves, especially at
the local scale. Although the NH simulations show a rapid damping, localized amplifications can be observed in more distant
coastal stretches, such as around Pozzuoli (northern coast) and the Nisida peninsula (on the east).
Figure 10 depicts the maximum tsunami amplitude along the coast vs the cumulative coastal distance, measured from the
eastern extreme of the computational domain and represented with the black labels, in the left plot. The SW-NH simulations
are almost indistinguishable for the three submarine scenarios (right plot in Figure 10), with the respective waves amplitude
that are almost superimposed. Cases 1 and 2 show limited effects on the coast, while Case 3 generates maximum amplitudes
of over 0.5 m between Pozzuoli and Nisida. As previously observed, on the contrary, for the subaerial case (Scenario 4) the
dispersive effects play a key role, lowering considerably the maximum amplitude at the coast starting from the Pozzuoli coastal
stretch (around 20 km of cumulative distance along the coast), with values almost halved at the opposite side of the Gulf.
Conversely, the two approaches produce similar waves for coastal stretches closer to the source, since for these the tsunami
travels in shallower water and the dispersive effects are then less impacting. As already observed, this scenario produces the
most impacting tsunami, with peak value close to 5 m in Capo Miseno and local amplifications at Pozzuoli and Nisida with
amplitudes of almost 2 m.

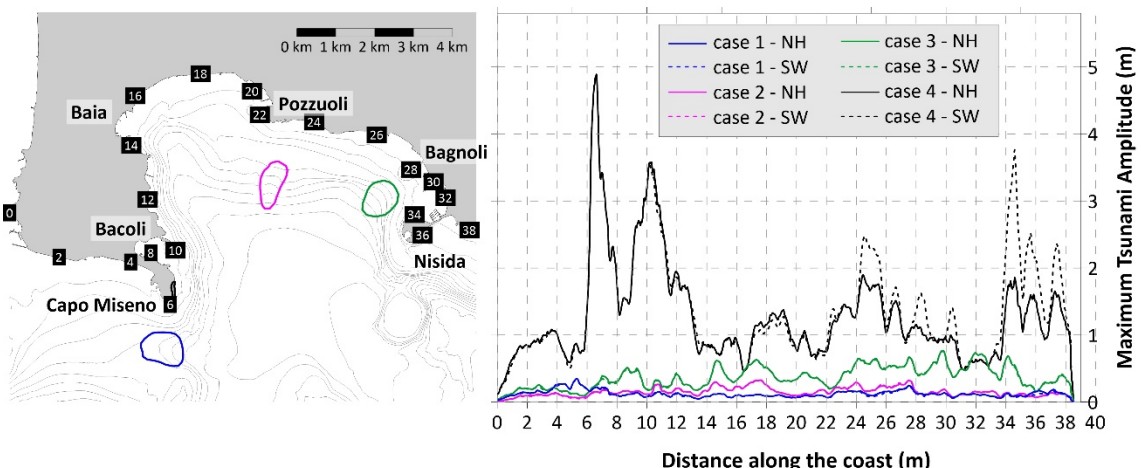


**Figure 10. The map on the left reports the initial boundaries of the landslide scenarios. The cumulative distance along the coast is**
**measured from the eastern extreme of the computational domain. The plot on the right depicts the maximum water amplitude along**
**the coast for the four scenarios, with comparison between NH (continuous lines) and SW (dashed lines).**




Figure 11 depicts the sketches of the first minutes of propagation for each scenario: all simulations show the typical feature
characterizing landslide-tsunamis, the almost circular shape of the tsunami signal, mimicking a point-like source. This polar
symmetry is lost when the wave interacts with shallow water and non-linear effects become predominant. A positive front
(yellow-red scale, meaning sea level increase) propagates in the same direction of the sliding motion, while a negative one
(cyan-blue scale) moves in the opposite side. For cases 2 and 3, the first manifestation of the tsunami at the coastal stretch
close to the source - presumably affected by the larger waves - is a negative signal, meaning that the water withdraws and is
followed later by a sea level increase: in terms of early warning this is an undoubted advantage, since it can act as a precursor
of an upcoming flooding. For case 1 the situation is different, since the slide motion does not have a direction opposite to the
dryland: a positive front enters the Gulf of Pozzuoli, while a negative one moves west of Capo Miseno, to the coastal stretch
out of the basin. As to case 4, the sliding motion starts in the subaerial environment, resulting into an always positive tsunami
front. Notice also the sequence of high frequency waves characterizing this scenario, especially evident in the 3- and 4-minutes
sketches (last row of plots), reflecting the smaller spatial dimensions of the tsunami source.

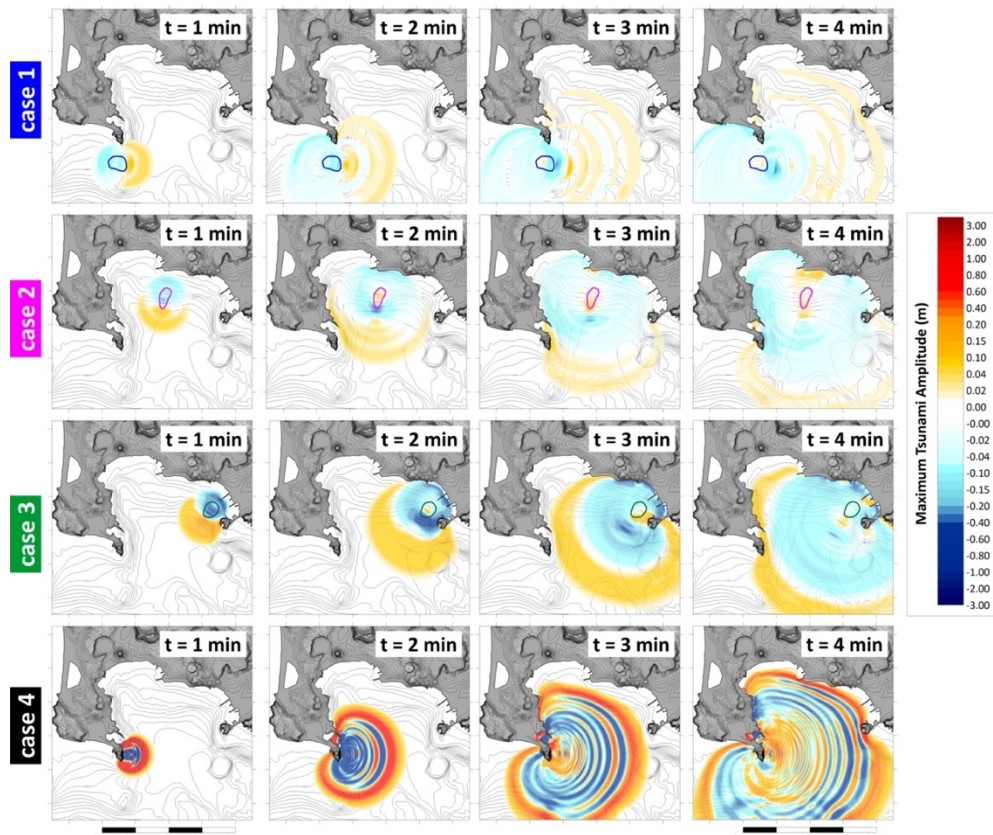


**Figure 11. Propagation sketches at 1 minute intervals of the four landslide-tsunami scenarios investigated (NH approach). The yellow-red scale marks the positive values (sea level increase), the cyan-blue scale is for negative ones (sea level sinking). The coloured contours represent the respective initial landslide boundary.**





In all cases represented in Figure 11, the tsunami affects most of the Gulf of Pozzuoli coasts within 4 minutes. Figure 12 reports
the travel time for each point of the computational domain, providing precious insights from the warning point of view:
independently from the initial position, the waves take approximately 3 to 6 minutes to affect all the coasts of the basin. Case
2 is the fastest, also due to its position at the centre of the Gulf. The north-western coastal stretch, on the contrary, is the one
reached latest (5 to 6 minutes) in every scenario: the waves are slowed down by the shallow-water shelf that in this area is
particularly large, compared to the other areas (as confirmed by the morphology, Figure 4). It is worth to specify that the code
JAGURS registers the first positive signal for each computational cell: this explains the anomalous pattern of the tsunami
opposite to the slide direction, particularly evident in Case 1 (westward, upper left panel, Figure 12) and Case 2 (northward,
upper right panel).

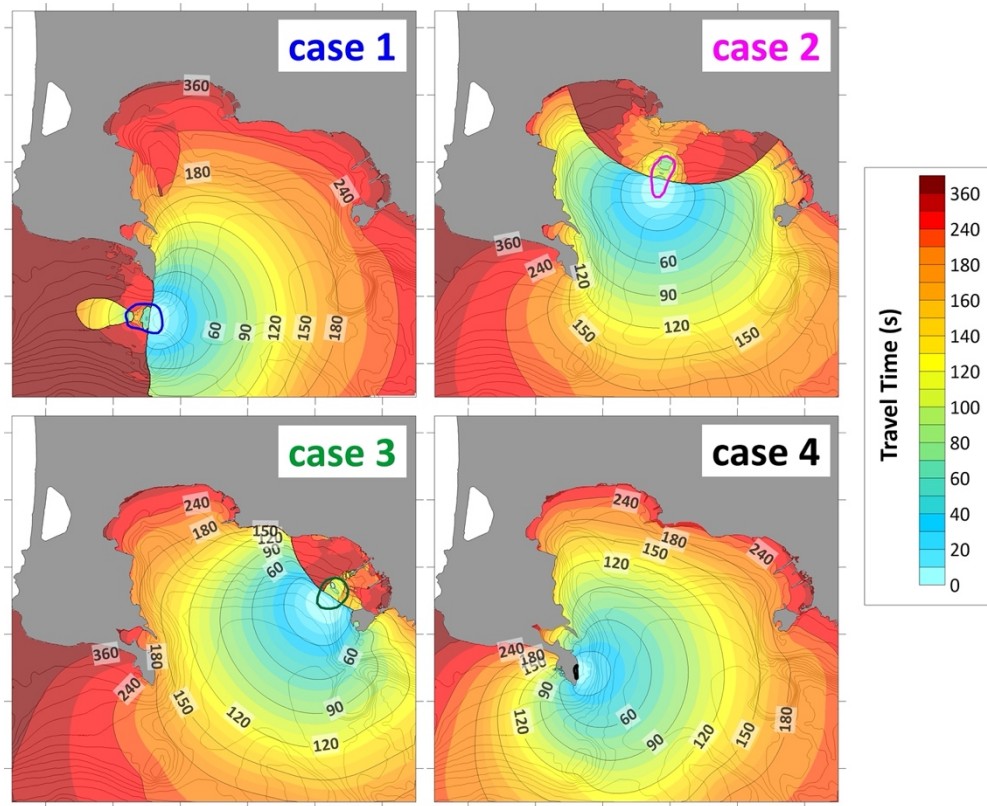


**Figure 12. Travel time, in seconds, of the tsunamis generated by the four landslide scenarios here hypothesized. The position of the**
**initial mass is marked by the respective coloured boundaries.**
**4 Discussion**
The numerical simulations illustrated in the previous section provide some precious insights about the tsunami hazard pattern
within the Gulf of Pozzuoli. Submarine collapses of the size adopted in this study generate waves that do not represent a threat
for coastal population, but which can damage small boats, which are present in a great number within the Gulf of Pozzuoli.





Conversely, the subaerial scenario produces high waves especially in the near field (almost 5 meters high), which rapidly
attenuate with distance thanks to dispersive effects. Some distant coastal stretches are affected by local amplifications, with
maximum amplitudes reaching almost 2 m (Pozzuoli, Nisida), highlighting the need to investigate this type of events. In most
cases, the tsunami reaches the shoreline in a few minutes with a positive signal, meaning that it manifests as a water level
increase. Only in limited coastal stretches, and not for all scenarios, the first signal is negative, i.e. the sea withdraws for some
minutes, providing a crucial precursor of an incoming wave in terms of early warning. In a few words, these events can occur
totally unannounced, reflecting the definition of "surprise tsunamis" given in Ward (2001). In the following, some specific
issues arising from the approach adopted and the simulation results are discussed.
*Landslide scenarios.* The scenario approach here adopted is a consequence of the lack of knowledge about the underwater
landslide bodies in the Gulf of Pozzuoli. Geophysical and bathymetric surveys have evidenced the presence of some ancient
collapses, buried by the sediments, in the deeper part of the basin at the toe of the slopes, but a general pattern of mass transport
processes in the Gulf of Pozzuoli, with recurrence time and volume estimation, doesn't exist. The sources adopted have been
reconstructed based on the pieces of evidence found in the scientific literature and on geomorphological considerations (margin
slopes, existing scars, basin depth), assuming that they are representative of the maximum credible occurrences expected in
the area. As a result, three submarine masses have been reconstructed, with similar volumes (few million m$^3$), thickness (a
maximum of about ten meters) and detachment depth (in shallow water, between 30 and 50 m). The fourth case is a coastal
subaerial collapse and is characterized by a very different morphology: it can be considered as the endmember of this type of
landslide, since no direct evidence exists of bigger collapses interacting with the sea. These scenarios cover the whole extent
of the Gulf, providing then a general idea of the impact expected from the ensuing tsunamis. However, larger collapses can't
be ruled out, especially in case of intensification of the Campi Flegrei volcanic activity providing possible triggers and, in the
long term, slope oversteepening.
*Dispersion effects.* For tsunamis of non-seismic origin, the effect of dispersion should be always taken into consideration, since
it can change consistently the propagation pattern with respect to the classic SW approach. The discrepancy grows with the
distance from the source, depending on the wavelength of the initial signal and on the depth of the basin where the perturbation
propagates. Eq. (1) provided rough estimates of this distance for the scenarios considered (reported in Table 4), suggesting
that for the submarine cases the dispersion is negligible within the Gulf of Pozzuoli domain. Numerical simulations confirmed
this hypothesis: the application of the code with (NH) and without (SW) dispersion produced almost identical tsunamis,
proving that the simpler and faster approach is sufficient to assess properly the tsunami hazard for these cases. For the subaerial
case, on the contrary, the difference is marked, as evidenced in Figure 10 by the maximum amplitudes along the coast. The
modelling effort, then, should consider the morphological features of the source generating the tsunami, keeping in mind that
subaerial masses collapsing into the sea usually generate shorter perturbations. In these cases, dispersive effects can become
relevant even for brief distances, and the application of the SW approach could produce an overestimation of the tsunami
impact on the coasts.





*Coastal and non-linear phenomena.* From the propagation plots (Figure 11) it is possible to infer some peculiar features of the
tsunami close to the coast, where the interaction with shallow water and minor basins can induce non-linear effects. For
example, for case 2 a positive signal propagating in the Bacoli Bay - a minor inlet just north of Capo Miseno on the western
part of the Gulf (see also Figure 13 for location) - can be noticed, evident especially in the $t = 3$ and 4 minutes sketches. This
is visible, while less marked, also for the other scenarios, and suggests the possibility of the excitation of the normal modes of
this sub-basin by the tsunami: this phenomenon, known as resonance, can occur for every basin affected by an external
perturbation, and is for example the mechanism at the basis of the generation of the meteotsunamis (Vilibic et al., 2016). The
morphology of the basin determines the periods of the resonant modes typical of the basin. Rabinovich (2009) obtained a set
of simple expressions allowing to estimate them for basic geometries. For example, for a rectangular basin of length $L$ and
uniform depth $h$, the period $T$ of the fundamental mode is:
$$T = \frac{2L}{\sqrt{gh}} \qquad\qquad\qquad\qquad (2)$$
where $g$ is the gravitational acceleration. Focusing on the Bacoli Bay case, Figure 13 reports the marigrams obtained from two
virtual tide gauges placed inside the inlet (tg#1, in red) and at its mouth (tg#2, in black), depicting the two respective time
histories in the four scenarios considered. The comparison of the signals shows clearly, for Cases 1, 2 and 3, that inside the
bay the perturbations behave as standing waves, characterized by regular oscillations lasting for at least 30 minutes (final
simulation time) with an approximate period of 200 s and evident amplifications if compared to the oscillations out of the basin
(in black). The tsunami generated by Case 3, in particular, is amplified five times with respect to the incoming signal. Assuming
for the Bacoli Bay a simplified rectangular geometry $(L \approx 1000 \ m, \ h \approx 10 \ m)$ and applying Eq. (2), one can estimate the
fundamental mode as $T \approx 200 \ s$, in full agreement with the features deduced from the virtual tide gauges. Moreover, the
subaerial scenario (Case 4) is less subject to the amplification when entering the inlet compared to the other cases, due to the
shorter period characterizing its oscillations.





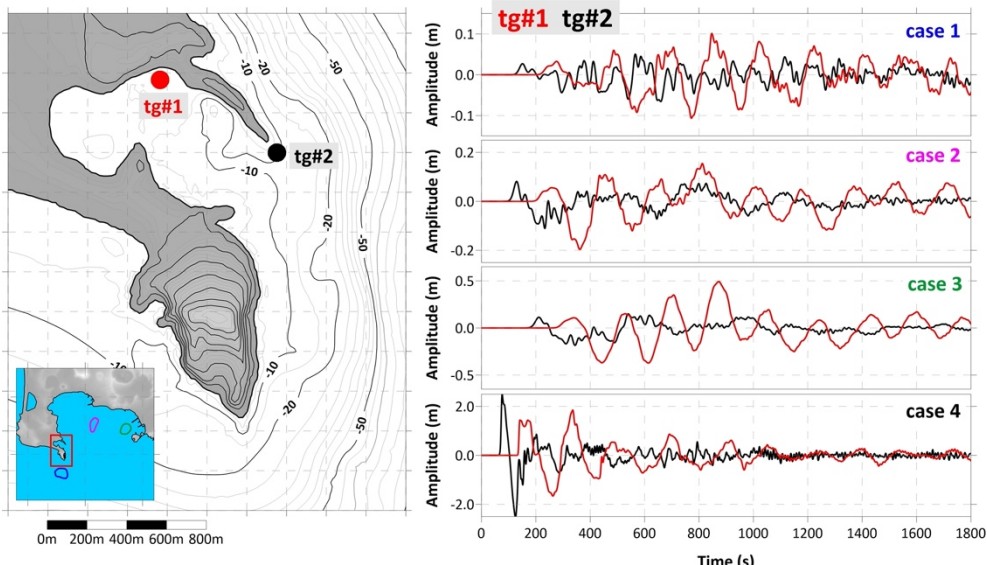

**Figure 13. Virtual tide gauges at the entrance (in black) and inside (in red) the Bacoli Bay. On the left, their location; on the right, the comparison between the marigrams for the scenarios here simulated.**

## 5 Conclusions

The Gulf of Pozzuoli covers a significant portion of the Campi Flegrei caldera, a region of significant geological and volcanic activity. Intense seismic and volcanic processes have the potential to destabilize large sliding bodies both in the subaerial and in the submarine environment. When these masses interact with the water, they can generate tsunamis that may impact the entire basin's coastline, posing potential risks to local communities, coastal infrastructures, and marine activities.

Despite the region being the object of extensive geological and geophysical investigations, a comprehensive understanding of mass transport processes remains limited. A scenario-based approach is then adopted, analysing four representative cases, based on the limited geological evidence and on morphological considerations: three submarine bodies, with similar volume (few million $m^3$, occurring in shallow water) and one subaerial slide (smaller mass, occurring on the coastline). These scenarios, though not reproducing actual occurrences, provide precious insights into the potential tsunami generation mechanisms and their subsequent impact on coastal areas, providing an important basis for possible risk mitigation strategies. The sliding dynamics and the resulting tsunamis are simulated using numerical techniques that account for key hydrodynamic phenomena such as dispersion, nonlinear coastal effects, and resonance.

The simulations results indicate that submarine landslides generally produce waves of limited amplitude on the coast, with a maximum height of 0.5 meters, as observed for case 3. While these tsunamis do not represent a major threat to coastal communities, they could still cause localized damages, for example to small boats moored in the harbours across the Gulf of Pozzuoli. Furthermore, resonance effects in smaller basins, such as harbours, can amplify an incoming wave, preventing its dissipation and resulting in standing wave affecting the coast for long time: simulation outcomes show that in the Bacoli Bay



- placed in the western sector of the gulf – the incoming signal is amplified up to five times, with a sequence of regular
oscillations with a period of 200 s, affecting the basin for at least 30 minutes, which reflects the normal modes typical of the
inlet. This can repeat in every coastal basin, each one characterized by its own geometry and fundamental mode and is worthy
of specific and detailed investigation. In contrast, the subaerial slide scenario results in significantly larger waves, exceeding
4 meters close to the source. The tsunami amplitude dampens rapidly with distance, due to dispersion, but in some coastal
stretches, Pozzuoli on the north and Nisida on the east, it reaches almost 2 meters.
The investigation here presented highlights the complex interplay between geological processes, hydrodynamic phenomena,
and coastal hazard in the Gulf of Pozzuoli. The results emphasize the need for detailed study and monitoring of potential
unstable masses, especially in coastal subaerial environment where they can give rise to large tsunamis threatening the whole
Gulf of Pozzuoli, and for risk management strategies to mitigate the potential impact of tsunamis in this active volcanic region.
**Code availability**
The numerical codes UBO-BLOCK and UBO-TSUIMP for landslide dynamics and tsunamigenic impulse computation
respectively are available under request to the authors; the code JAGURS can be freely downloaded from the link:
https://github.com/jagurs-admin/jagurs.
**Data availability**
The computational grids have been obtained from the elaboration of raw datasets available online, which have been
interpolated and readjusted. They are available under request to the authors.
**Author contribution**
FZ and AA conceptualized the investigation; FZ, CA and MZ prepared the computational grids and the landslide scenario
datasets; FZ and LS performed the simulations; FZ prepared the manuscript, with the contribution from all co-authors; FZ and
AA supervised the whole manuscript realization.
**Competing interests**
The authors declare that they have no conflict of interest.



**Acknowledgements**
The authors are grateful to Prof. Jacopo Selva, University of Naples, Federico II, for the productive discussion about the
potential tsunami sources in the Gulf of Pozzuoli related to the Campi Flegrei activity.

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
