# Peer review of "Tsunamigenic potential of unstable masses in the Gulf of Pozzuoli,"

_EGUsphere, 2025_

## Author Comment (AC1)

**Authors' comments - REVIEWER#1**

This study investigates the potential tsunami hazard from landslide scenarios in the Gulf of Pozzuoli using a sequence of numerical models. The authors present four scenarios (three submarine and one subaerial) and simulate the associated tsunamis using both shallow water (SW) and non-hydrostatic (NH) models. The study addresses important local hazard concerns in a densely populated coastal area. While the paper is generally well-structured, several critical aspects require clarification and improvement before the manuscript can be considered for publication.

We thank the reviewer for the insightful and constructive remarks, improving consistently the paper clearness and efficacy. In the following, the reviewer's comments are reported in black, and our replies and comments in red.

**Major Comments**

**Landslide Scenario Definition and Assumptions**

The tsunami waveforms generated by landslides are sensitive to the initial conditions, including the location, volume, geometry, and material properties of the sliding mass. The authors mention that the four scenarios were constructed based on a "worst-case credible" approach. However, the paper lacks sufficient detail on how the initial conditions, especially volume and geometry, were determined. The cited reference (Zaniboni and Armigliato, 2025) is listed as a work in progress and is not yet available, making it difficult to assess the robustness of the scenarios.

The citation refers to a book chapter that was supposed to be published last year. Unfortunately, it faced many difficulties in the editing process and now it still in progress. Another reference will be provided (Tonini et al., 2011), describing the approach and its application to the specific case of the town of Catania (Sicily, South Italy). However, we believe that the "worst-case credible" approach is not completely suitable for this investigation, due to the lack of detailed surveys mapping the Gulf of Pozzuoli seabed and to the rapidly changing morphology of the area: then it will be adopted and described, with its limitations and cautions properly addressed. As to the strategy of the scenario selection and definition, it has been outlined in the respective subsection in Discussion chapter.

Moreover, the assumption that geotechnical and geomorphic properties are similar across all four scenarios is not justified or even explicitly stated. This assumption should be clarified, as differences in material properties can significantly influence landslide dynamics and tsunami generation. For instance, variations in density, cohesion, yield strength and/or internal friction angle can lead to different failure mechanisms and velocities, thereby affecting the characteristics of the generated tsunami waves.

The authors should provide more detailed explanations for each scenario's setup, including volume estimates, slope angle, and material assumptions. If geotechnical properties are assumed identical across cases, this simplification should be clearly stated and discussed, along with a justification for why this simplification is reasonable in this context.

We agree that the landslide properties can influence deeply the tsunami generation, but the detailed reconstruction of the scenarios requires detailed surveys and characterization of the deposits. Such tasks are far beyond the purposes of this work which, indeed, could stimulate further investigations in the area. However, some of these considerations will be added in the landslide scenario sub-section of the Discussion and in the Conclusions.

**Tsunami Generation Mechanism and Modeling Approach**

The modeling approach uses a one-way coupling scheme, where landslide motion is simulated independently and used as input for tsunami generation and propagation. This approach, while computationally efficient, may be insufficient to capture certain physical mechanisms, particularly for subaerial landslides like scenario 4, where the interaction with the water column is highly dynamic and nonlinear. In reality, the water displaced by the landslide can, in turn, influence the landslide's motion.

This is true, but for example in Harbitz et al. (2006) it is stated that, based also on previous studies, the velocity of the landslide front could be reduced of up to 20% by the interaction with the wave it generates. Though not irrelevant, this effect does not affect consistently the generated tsunami and its propagation in the Gulf of Pozzuoli, which is the main focus of this work.

The subaerial case involves a mass plunging from above sea level into the water with high velocity, which contrasts with the more gradual submarine slope failures of the other scenarios. Given these differences in physical processes, it is unclear whether the same numerical treatment is equally valid for both types of landslide.

The two landslide motions are quite different, but the mechanism of tsunami generation is the same, i.e. the uplift of the whole water column due to the passage of the sliding mass on the seabed. Other highly non-linear processes occur in the landslide-water interaction for subaerial cases, but they usually generate effects only in the near field, dissipating quickly with distance from the crash area. In this view, we believe that the modelling approach is suitable for both typology of movement. We'll add some references to cases of subaerial landslides producing tsunamis where our approach produced very good results (Scilla 1783 landslide-tsunami, for example).

The authors should justify the use of the same modeling framework across all scenarios and clearly discuss the limitations of their approach, especially regarding the subaerial case. They should acknowledge the potential limitations of the one-way coupling and discuss how this might affect the accuracy of their results.

Some sentences on this regard will be added in the section describing the modelling approach and in the Discussion section.

**Role of Froude Number and Energy Transfer Efficiency**

A crucial omission in the discussion of tsunami generation is the role of the Froude number Fr=U/gH, which characterizes the relationship between the landslide velocity (U) and the shallow water wave speed (gH). When the Froude number approaches 1, energy transfer from the landslide to the water is most efficient due to resonance-like effects. This can lead to significant amplification of the generated waves.

While the authors analyze landslide velocities and acknowledge the influence of slide speed and dispersion, they do not discuss the possible amplification effects that occur when the slide velocity approaches the wave celerity. This is particularly relevant for Scenario 4.

The authors should discuss whether any of their scenarios reach near-critical Froude conditions and, if so, whether their model can appropriately represent the associated amplification. Even a simplified estimation of the Froude number for each case would enhance the paper. This analysis would provide a more complete understanding of the tsunami generation process and its potential impact.

We totally agree with this view. The Froude number (defined as $U/\sqrt{gh}$) is a precious indicator of the efficiency of the energy transfer from the mass to the water. We will add its estimation in the plots for each scenario and discuss it accordingly.

**Minor Comments:**

L9 "consisting into" -> "consisting of"

Ok

L38 "In the specific" -> "Specifically"

Ok.

L43 "assessing" -> "by assessing"

Ok.

L128 "with no back-interactions considered" -> "and back-interactions are not considered"

Ok.

L134 "a finite time" ->unclear since an earthquake occurs in a finite time. Consider rephrasing to something like "a non-instantaneous generation process" or "a generation process that evolves over time".

Ok.

Table 2 - I do not think Table 2 is necessary. We may replace it by a paragraph. Also Table 2 was wrongly referred to at L284 and L308. They should be Table 3 instead.

We agree, Table 2 will be removed, and numbering and references adjusted accordingly.

Figure 1&2 - Two figures can be combined.

We prefer to leave them separated: Fig.1 gives a general overview of the area, including the whole Gulf of Naples, while Fig.2 focuses on the area object of the study, with some toponyms that will be cited later and that wouldn't be discernable in a larger map.

Figure 3 is almost identical to Figure 2&4 of Aiello et al. (2012). I am concerned about the permission to use these figures. The authors should provide confirmation that they have obtained the necessary permissions to reproduce or adapt these figures.

Figure 3 will be designed from scratch, based on the plots in Aiello et al (2012), to avoid copyright issues.

---

## Author Comment (AC2)

**Replies to REVIEWER#2**

The manuscript titled "Tsunamigenic potential of unstable masses in the Gulf of Pozzuoli, Campi Flegrei, Italy" provides a numerical study of four submarine (three) and subaerial (one) landslide scenarios and their corresponding tsunamigenic impact in the Gulf of Pozzuoli. The work is looking at past evidence of collapse to reproduce modes of failure and tsunamigenesis with the largest potential attributed to subaerial collapse. The codes UBO-BLOCK and UBO-TSUIMP/JAGURS are used for simulating mass failure and tsunami respectively. The dispersive potential of the wave characteristics is also investigated in the study. The authors conclude that only some of the scenarios examined have an impact on the adjacent coastlines. This is a comprehensive investigative work that can strengthen the existing knowledge of tsunamis in the region. However, some key points need to be addressed.

We thank the reviewer for the interesting and productive comments, that will help to improve this work. Specific answers to the remarks are reported below in red.

**General comments**

1) P.2 L45-46, P8 L196-198, P17 L355 and elsewhere

Although it is noted that the masses have been selected based on previous deposits and present morphology, it is not clear why these are the worst possible scenarios that could happen in the region. Can the authors affirm this with certainty? This needs to be carefully addressed as it can be misleading for future policy and hazard mitigation efforts as the current work indicates that there is not a significant risk from submarine but only subaerial landslide tsunamis in the region. Future failures may not replicate past events and an increase in volcanic activity, variance in the location of failure, and higher collapse volumes may increase tsunamigenic potential and the impact may be larger, even more so if the possibility of such events is ruled out.

We totally agree with this view. The cited "worst-case credible" approach bases on the known and potential sources, or on credible scenarios built on the existing evidence. Due to the poor knowledge of the seabed morphology and to the rapidly changing conditions of the area, mobilization of bigger volumes can't be ruled out. We will add these considerations when citing the worst-case credible approach, evidencing its limitations in this case.

2) S2.2

Subaerial landslide tsunamis generally have a greater tsunamigenic potential than submarine ones and have more complexity due to the interactions between the solid mass, water and air. Modelling the high impact, and complex slide kinematics often requires 3D Navier-Stokes solvers, SPH, CFD or VOF models. It is not clear whether this complexity is captured with the study's numerical codes, an issue which could underestimate the hazard. It is not clarified in the manuscript how the authors distinguish between the modelling of the submarine and subaerial failure, besides the different solvers used in the propagation modelling.

The impact of the sliding mass with water is a very complex process, involving highly non-linear phenomena that are hard to capture with numerical simulations. At the same time, they are mainly confined to the "splash-zone", the area around the impact which has typically limited extension. Moreover, the generated waves can be very high but have usually high frequency, then dissipate quickly. The perturbation propagating at higher distance in the Gulf of Pozzuoli, on the contrary, results from the longer components, which can be associated to the landslide motion along the submarine portion of the sliding surface. This is

adequately captured, in our view, from our modeling approach. Some citations of cases of subaerial landslide-tsunami for which our approach produced very good results will be added (for example, Scilla 1783 landslide-tsunami).

**Specific comments:**

P.2 L45-46 The worst-case referenced approach is not yet published and therefore hard to understand and verify. The authors should expand more on the methodology.

As stated above, the reference to the worst-case will be removed.

P.2 L127, 128-130: Air Entrainment is of importance when focusing on subaerial landslides, the generation involves a triple phase interaction.

We will adjust the sentences, even though in our approach we will not model such interaction.

P3 L139-141. What are the underpinning equations in UBO-BLOCK? The model seems to have been primarily used in the modelling of submarine landslide tsunamis

Indeed, it has been tested also in the subaerial environment, for example in the 1783 Scilla landslide-tsunami, or in the 1963 Vajont landslide. References to these two cases will be added.

P6 L167 As JAGURS is nested, it is worth clarifying that the nested approach is omitted in that case.

Ok

P6 L167 Please also give more details on whether the simulations were run locally or in a cluster, CPU time and time of the event.

The simulations are run on local computers, with order of magnitude of ten of minutes for the landslide dynamics and some hours for the tsunami propagation. In this last case, JAGURS is not used with the parallel computing version, so the computational time is not optimized, then not particularly significant.

P10 L231-233 This statement should be substantiated by references.

Ok

P10 L240-254 Although the authors mention the locations of the deposits and the corresponding volumes it is not clear how these volumes were estimated based on the observed deposits. A few sentences explaining the approach would help here.

The approach used to reconstruct the scenarios is described more extensively in the Discussion section, under the paragraph *Landslide scenarios*. We will add some sentences on this regard also in the scenario description.

P11 L263-265 If any, what kind of rheology is assumed for the sliding mass?

Due to the few information available for the area, we did not describe in detail the landslide rheology. We assumed for them a moderate translational behavior, based on the scarce information available: for example, for Scenario 1 the position of the final deposit is estimated, and this is assumed as a constraint on the dynamics and the rheology. This is not the case for the other scenarios, for which a similar behavior has been assumed. Of course, there is a big uncertainty on the landslide rheology and geometry: this work aims also at stimulating further investigation in the area. Some of these considerations will be added in the scenario description and in the conclusions.

P17 L355 This statement standing alone reads quite strong and I think it cannot be backed up by only assessing 3 scenarios of collapse, it should be better rephrased to the specific case studies as in P19 L408.

Agree, the sentence will be modified.

Figure 10 I think xlabel is mistyped and '(m)' is wrong?

Yes, thanks for the observation, we'll fix it.

P20 L427-429 I agree with the authors, and I think this is not clear throughout the manuscript so far.

The discussion about the scenario individuation has been included in the Discussion session to avoid repetitions and group all the criticalities. In our view, in this way the paper is clearer and better organized. However, some words about this will be added also in the scenario description.

P22 L472-L479 I believe an important addition to the discussion would be how parameterisation and probabilistic approaches, as for example surrogate models, can help when it comes to future hazard assessments in the region and the capability of assessing multiple scenarios of collapse with large variances in location, volume and mode of failure.

This is an interesting point, we will add some considerations about this in the conclusions.

---

## Author Response (AR1)

**Authors' comments - REVIEWER#1**

This study investigates the potential tsunami hazard from landslide scenarios in the Gulf of Pozzuoli using a sequence of numerical models. The authors present four scenarios (three submarine and one subaerial) and simulate the associated tsunamis using both shallow water (SW) and non-hydrostatic (NH) models. The study addresses important local hazard concerns in a densely populated coastal area. While the paper is generally well-structured, several critical aspects require clarification and improvement before the manuscript can be considered for publication.

We thank the reviewer for the insightful and constructive remarks, improving consistently the paper clearness and efficacy. In the following, the reviewer's comments are reported in black, and our replies and comments in red.

**Major Comments**

**Landslide Scenario Definition and Assumptions**

The tsunami waveforms generated by landslides are sensitive to the initial conditions, including the location, volume, geometry, and material properties of the sliding mass. The authors mention that the four scenarios were constructed based on a "worst-case credible" approach. However, the paper lacks sufficient detail on how the initial conditions, especially volume and geometry, were determined. The cited reference (Zaniboni and Armigliato, 2025) is listed as a work in progress and is not yet available, making it difficult to assess the robustness of the scenarios.

We maintained the worst-case credible approach describing its limitations due to the lack of surveys mapping the Gulf of Pozzuoli seabed and to the rapidly changing morphology of the area (see Paragraph 2.4 and some small modifications in Discussion and Conclusions sections). The citation refers to a book chapter that was supposed to be published within 2024. Unfortunately, it is facing editorial difficulties and now is still in progress (currently, the supposed publication date is December 2025, so we hope to have it published soon). An additional reference has been provided (Tonini et al., 2011), describing the same approach and its application to the specific case of the town of Catania (Sicily, South Italy).

Moreover, the assumption that geotechnical and geomorphic properties are similar across all four scenarios is not justified or even explicitly stated. This assumption should be clarified, as differences in material properties can significantly influence landslide dynamics and tsunami generation. For instance, variations in density, cohesion, yield strength and/or internal friction angle can lead to different failure mechanisms and velocities, thereby affecting the characteristics of the generated tsunami waves. The authors should provide more detailed explanations for each scenario's setup, including volume estimates, slope angle, and material assumptions. If geotechnical properties are assumed identical across cases, this simplification should be clearly stated and discussed, along with a justification for why this simplification is reasonable in this context.

We agree that the landslide properties can influence deeply the tsunami generation, but the detailed reconstruction of the scenarios requires detailed surveys and characterization of the deposits. Such tasks are far beyond the purposes of this work which, indeed, could stimulate further investigations in the area (as suggested in the Conclusions). However, the scenarios description has been slightly improved and moved to Data and Methods section, and a short discussion about this has been added at the end of Paragraph 2.4 and resumed in Discussion and Conclusion sections.

**Tsunami Generation Mechanism and Modeling Approach**

The modeling approach uses a one-way coupling scheme, where landslide motion is simulated independently and used as input for tsunami generation and propagation. This approach, while computationally efficient, may be insufficient to capture certain physical mechanisms, particularly for subaerial landslides like scenario 4, where the interaction with the water column is highly dynamic and nonlinear. In reality, the water displaced by the landslide can, in turn, influence the landslide's motion.

This is true, but for example in Harbitz et al. (2006) it is stated that, based also on previous studies, the velocity of the landslide front could be reduced of up to 20% by the interaction with the wave it generates. Though not irrelevant, this effect does not affect consistently the generated tsunami and its propagation in the Gulf of Pozzuoli, which is the main focus of this work. These considerations were already reported in Paragraph 2.2, and have been slightly modified adding this citation.

The subaerial case involves a mass plunging from above sea level into the water with high velocity, which contrasts with the more gradual submarine slope failures of the other scenarios. Given these differences in physical processes, it is unclear whether the same numerical treatment is equally valid for both types of landslide. The authors should justify the use of the same modeling framework across all scenarios and clearly discuss the limitations of their approach, especially regarding the subaerial case. They should acknowledge the potential limitations of the one-way coupling and discuss how this might affect the accuracy of their results.

The mechanism of tsunami generation remains the same, i.e. the uplift of the whole water column caused by the passage of the sliding mass along the seabed. Other highly non-linear processes occur in the landslide-water interaction in subaerial cases, but these effects are generally confined only to the near field and dissipate rapidly with distance from the impact area. In this view, we believe that the modelling approach is suitable for both types of movement. We added a subsection in Discussion specifically addressing this issue.

**Role of Froude Number and Energy Transfer Efficiency**

A crucial omission in the discussion of tsunami generation is the role of the Froude number Fr=U/gH, which characterizes the relationship between the landslide velocity (U) and the shallow water wave speed (gH). When the Froude number approaches 1, energy transfer from the landslide to the water is most efficient due to resonance-like effects. This can lead to significant amplification of the generated waves.

While the authors analyze landslide velocities and acknowledge the influence of slide speed and dispersion, they do not discuss the possible amplification effects that occur when the slide velocity approaches the wave celerity. This is particularly relevant for Scenario 4.

The authors should discuss whether any of their scenarios reach near-critical Froude conditions and, if so, whether their model can appropriately represent the associated amplification. Even a simplified estimation of the Froude number for each case would enhance the paper. This analysis would provide a more complete understanding of the tsunami generation process and its potential impact.

We totally agree with this view. The Froude number (defined as $U/\sqrt{gh}$) is a precious indicator of the efficiency of the energy transfer from the mass to the water. It has been introduced and described in Paragraph 3.1.1 (for Scenario 1) and added to the plots and commented for each scenario.

**Minor Comments:**

L9 "consisting into" -> "consisting of"

Ok

L38 "In the specific" -> "Specifically"

Ok.

L43 "assessing" -> "by assessing"

Ok.

L128 "with no back-interactions considered" -> "and back-interactions are not considered"

Ok.

L134 "a finite time" ->unclear since an earthquake occurs in a finite time. Consider rephrasing to something like "a non-instantaneous generation process" or "a generation process that evolves over time".

Ok.

Table 2 - I do not think Table 2 is necessary. We may replace it by a paragraph. Also Table 2 was wrongly referred to at L284 and L308. They should be Table 3 instead.

We agree, Table 2 will be removed, and numbering and references adjusted accordingly.

Figure 1&2 - Two figures can be combined.

Figure 2 has been removed, and numbering changes accordingly.

Figure 3 is almost identical to Figure 2&4 of Aiello et al. (2012). I am concerned about the permission to use these figures. The authors should provide confirmation that they have obtained the necessary permissions to reproduce or adapt these figures.

Figure 3 (renamed as Figure 2 in the new version) was created entirely from scratch, based on the plots in Aiello et al (2012), to avoid copyright infringement.

The manuscript titled "Tsunamigenic potential of unstable masses in the Gulf of Pozzuoli, Campi Flegrei, Italy" provides a numerical study of four submarine (three) and subaerial (one) landslide scenarios and their corresponding tsunamigenic impact in the Gulf of Pozzuoli. The work is looking at past evidence of collapse to reproduce modes of failure and tsunamigenesis with the largest potential attributed to subaerial collapse. The codes UBO-BLOCK and UBO-TSUIMP/JAGURS are used for simulating mass failure and tsunami respectively. The dispersive potential of the wave characteristics is also investigated in the study. The authors conclude that only some of the scenarios examined have an impact on the adjacent coastlines. This is a comprehensive investigative work that can strengthen the existing knowledge of tsunamis in the region. However, some key points need to be addressed.

We thank the reviewer for the interesting and productive comments, that help to improve this work. Specific answers to the remarks are reported below in red.

**General comments**

1) P.2 L45-46, P8 L196-198, P17 L355 and elsewhere

Although it is noted that the masses have been selected based on previous deposits and present morphology, it is not clear why these are the worst possible scenarios that could happen in the region. Can the authors affirm this with certainty? This needs to be carefully addressed as it can be misleading for future policy and hazard mitigation efforts as the current work indicates that there is not a significant risk from submarine but only subaerial landslide tsunamis in the region. Future failures may not replicate past events and an increase in volcanic activity, variance in the location of failure, and higher collapse volumes may increase tsunamigenic potential and the impact may be larger, even more so if the possibility of such events is ruled out.

We totally agree with this view. The cited "worst-case credible" approach is based on the known and potential sources, or on credible scenarios built on the existing evidence. Due to the poor knowledge of the seabed morphology and to the rapidly changing conditions of the area, the mobilization of larger volumes can't be ruled out. We added these considerations in the description of the worst-case credible approach (Paragraph 2.4), highlightinging its limitations in this case.

2) S2.2

Subaerial landslide tsunamis generally have a greater tsunamigenic potential than submarine ones and have more complexity due to the interactions between the solid mass, water and air. Modelling the high impact, and complex slide kinematics often requires 3D Navier-Stokes solvers, SPH, CFD or VOF models. It is not clear whether this complexity is captured with the study's numerical codes, an issue which could underestimate the hazard. It is not clarified in the manuscript how the authors distinguish between the modelling of the submarine and subaerial failure, besides the different solvers used in the propagation modelling.

The impact of the sliding mass on water is a very complex process, involving strongly non-linear phenomena that are hard to capture with numerical simulations. At the same time, these effects are mainly confined to the "splash-zone", the area around the impact that typically has limited extent. Moreover, the generated waves can be very high but usually have high frequencies, and hence dissipate quickly. In contrast, the perturbation propagating to greater distances in the Gulf of Pozzuoli results from the longer wave components, which can be associated with the landslide motion along the submarine portion of the

sliding surface. In our view, this behaviour is adequately captured from our modeling approach. These considerations have been included in a specific and dedicated subsection of the Discussion chapter.

**Specific comments:**

P.2 L45-46 The worst-case referenced approach is not yet published and therefore hard to understand and verify. The authors should expand more on the methodology.

An additional reference to a previous paper (Tonini et al., 2011), describing the same approach and its application, has been added. The 2025 chapter is in phase of publication and we hope to have it available soon, however its citation will be removed.

P.2 L127, 128-130: Air Entrainment is of importance when focusing on subaerial landslides, the generation involves a triple phase interaction.

As already mentioned, from the point of view of the tsunami propagation in the Gulf such effect is secondary, and limited to the source area.

P3 L139-141. What are the underpinning equations in UBO-BLOCK? The model seems to have been primarily used in the modelling of submarine landslide tsunamis

Indeed, it has been tested also in the subaerial environment, for example in the 1783 Scilla landslide-tsunami. References to this case have been added.

P6 L167 As JAGURS is nested, it is worth clarifying that the nested approach is omitted in that case.

Ok, done (Paragraph 3.2).

P6 L167 Please also give more details on whether the simulations were run locally or in a cluster, CPU time and time of the event.

The simulations are run on local computers, with order of magnitude of ten of minutes for the landslide dynamics and some hours for the tsunami propagation. In this last case, JAGURS is not used with the parallel computing version: the computational time is not optimized, then not particularly significant.

P10 L231-233 This statement should be substantiated by references.

A reference to the June 30th, 2025 event has been added. However, the topic is discussed widely in other sections.

P10 L240-254 Although the authors mention the locations of the deposits and the corresponding volumes it is not clear how these volumes were estimated based on the observed deposits. A few sentences explaining the approach would help here.

The description of the scenario reconstruction has been slightly extended and moved to Paragraph 2.4.

P11 L263-265 If any, what kind of rheology is assumed for the sliding mass?

Given the limited information available for the area, we did not describe in detail the landslides' rheology. We assumed a moderate translational behavior, based on the scarce information available: for example, in Scenario 1 the position of the final deposit is estimated, and this is assumed as a constraint for both the dynamics and rheology. For the other scenarios this information is not available, but a similar behavior has been assumed. Of course, there is a considerable uncertainty regarding the landslide rheology and geometry: this work aims also at encouraging further investigation in the area. Some of these considerations have been added at the end of Paragraph 2.4 and resumed in Discussion and in Conclusions.

P17 L355 This statement standing alone reads quite strong and I think it cannot be backed up by only assessing 3 scenarios of collapse, it should be better rephrased to the specific case studies as in P19 L408.

Agree, the sentence has been modified.

Figure 10 I think xlabel is mistyped and '(m)' is wrong?

Yes, thanks for the observation, fixed.

P20 L427-429 I agree with the authors, and I think this is not clear throughout the manuscript so far.

The discussion about the scenario individuation has been moved to Paragraph 2.4 and more stressed, and resumed in the Discussion session.

P22 L472-L479 I believe an important addition to the discussion would be how parameterisation and probabilistic approaches, as for example surrogate models, can help when it comes to future hazard assessments in the region and the capability of assessing multiple scenarios of collapse with large variances in location, volume and mode of failure.

These are indeed interesting points, but we preferred not to raise such topics, which are still developing.

---

## Author Response (AR2)

Dear Editor,

we have read the reviewer's comments and went through the manuscript fixing and implementing most of the proposals and the remarks presented.

We don't agree with the main point raised, concerning the Froude number definition, and we didn't change the plots representing this parameter as proposed, since we believe that our computations are correct: please read our answer to 1. Major comment for more details.

The reviewer's remarks are reported in black; our replies are in red. Please consider that the line numbers cited in the answers refer to the new version of the manuscript, with track changes enabled.

Waiting for your feedback, we thank both the reviewer and you for your work.

Best regards

Prof. Filippo Zaniboni (corresponding author)

**Authors' comments - REVIEWER#1**

The authors have improved the manuscript. However, there are still a few points that should be addressed before publication. The most important one concerns the definition and interpretation of the Froude number used throughout the results. I also list minor comments.

1. Major comment – Froude number: definition and interpretation

In the revision, the Froude number is introduced as "the ratio between the landslide velocity and the tsunami phase velocity in the shallow-water approximation". The authors need to give the definition clearly. The standard definition is Fr=U/gH, where U is the speed of slide and H is the local water depth. In practice, U is the frontal speed of the landslide, and H is the water depth at the front. And thus Froude number cannot be defined when the slide plunges into the water.
The Froude number is a dimensionless ratio of velocities. Near-critical conditions (Fr ~ 1) indicate strong kinematic coupling between slide and long waves and are favourable for efficient energy transfer, but Fr does not itself "measure the efficiency of the energy transfer". Wave generation also depends on slide volume, thickness, density contrast, geometry, etc.
I think the Fr number was wrongly calculated, and thus the authors update these figures.

To be a dimensionless parameter, $Fr$ must be defined as we did in the text: the term $gh$ has the dimensions of a squared velocity, then the definition proposed by the reviewer implies a Froude number with dimensions $v^{-1}$. Indeed, the standard definition of the Froude number is $Fr = U/\sqrt{gh}$, with $U$ as the horizontal speed of the landslide and $\sqrt{gh}$ corresponding to the tsunami phase velocity in SW. In this version, $Fr$ becomes a velocity ratio, as the reviewer him/herself states. We did not change, then, the Froude number computations and the respective figures.

We agree with the comment that $Fr$ is not the only parameter influencing tsunami generation, even though that sentence was not intended to state this. However, we have changed the sentence to "$Fr$ provides an indication of the efficiency of the energy transfer" (L302), hoping this make it clearer.

**2. Minor comments**

L97–98 "A tsunami can be considered mostly as a packet of waves …"
This statement is vague and a bit misleading. Tsunami wave trains can indeed be described as packets of long waves with a characteristic period range, but this is not a universal or sufficiently precise definition.

We wanted to suggest the idea that a tsunami is not a monochromatic oscillation but incorporates several components with different period. The sentence has been slightly rearranged, removing the "packet of waves" term (L97).

L166 – "regularly"
The meaning of "regularly" is unclear here. If you mean "equally spaced" or "uniformly distributed", please use that wording. Otherwise, specify exactly what property is "regular": spacing, amplitude, temporal sampling, etc.

Fixed (L171).

L176–177 – "shows some remarkable features as concerns landslide-tsunami triggering and propagation"
This phrase is unclear.

The sentence has been rephrased (L181-182).

L447–452 – very long, unclear sentence
The sentence spanning L447–452 is too long and hard to parse. Please split it into two or three shorter sentences, each conveying a single main idea.

In the new version of the manuscript, such lines correspond to L457-462 where, indeed, there were already three separated sentences. We have split the first one into two shorter phrases (L458).

Table 3 – inconsistency in λ, b and λ/h
There is an inconsistency: in the text the initial wavelength is approximated as $\lambda \approx 2b$, while in the Table 3 caption states that λ coincides with b.

Fixed, thank you.

**3. References**
Some in-text citations are missing from the reference list, e.g. Ward (2001) ("surprise tsunamis") and Selva et al. (2018) (Introduction).

Reference to Ward (2001) added, Selva et al is (2021) and not (2018), and has been corrected in the Introduction.

Some references appear to have an incorrect year (e.g. Abadie et al., "La Houille Blanche …", where the DOI suggests a different year than the one printed). Please verify against the original journal.

Abadie et al year fixed, all citations checked.

Verify DOI/URL at the end of the bibliography that duplicate information already contained in properly formatted entries. These look like copy-paste artefacts from a reference manager and should be removed, keeping only the standard formatted references.

Checked and fixed.